# Optimization and inventory management under stochastic demand using metaheuristic algorithm

Nguyen Duy Tan[1], Hwan-Seong Kim[1]*, Le Ngoc Bao Long[1], Duy Anh Nguyen[2], Sam-Sang You[3]

1 Department of Logistics, Korea Maritime and Ocean University, Busan, Republic of Korea, 2 Department of Mechatronics, Ho Chi Minh City University of Technology (HCMUT)-Vietnam National University Ho Chi Minh City, Ho Chi Minh City, Vietnam, 3 Division of Mechanical Engineering, Northeast-Asia Shipping and Port Logistics Research Center, Korea Maritime and Ocean University, Busan, Republic of Korea

* kimhsyskmou@gmail.com

**Data Availability Statement:** All relevant data are provided within the paper.

**Funding:** This research was supported by Korea Institute of Marine Science & Technology

## Abstract

This study considers multi-period inventory systems for optimizing profit and storage space under stochastic demand. A nonlinear programming model based on random demand is proposed to simulate the inventory operation. The effective inventory management system is realized using a multi-objective grey wolf optimization (MOGWO) method, reducing storage space while maximizing profit. Numerical outcomes are used to confirm the efficacy of the optimal solutions. The numerical analysis and tests for multi-objective inventory optimization are performed in the four practical scenarios. The inventory model's sensitivity analysis is performed to verify the optimal solutions further. Especially the proposed approach allows businesses to optimize profits while regulating the storage space required to operate in inventory management. The supply chain performance can be significantly enhanced using inventory management strategies and inventory management practices. Finally, the novel decision-making strategy can offer new insights into effectively managing digital supply chain networks against market volatility.

## 1. Introduction

Over the past several decades, several inventory models have been an attractive topic of conversation in operations research, highlighting the distinctions between historical academic research and real-world implementations. While academic research focuses on employing mathematical control techniques for a few unique inventory models, industrial situations attempt to address practical issues by examining the maximal revenues or the minor expenses rather than balancing several objectives. Most inventory management issues are focused on a single goal, mainly using the classic inventory model, which defines an item as manufactured or acquired by management. However, those assumptions might not hold for a real test of business resilience and risk management in logistics and supply chain management. To engage in more lucrative operations, many businesses, enterprises, or vendors deal with various goods

Promotion (KIMST) funded by the Ministry of Oceans and Fisheries, Korea (20220573). The funders had no role in study design, data collection and analysis, decision to publish, or preparation of the manuscript.

**Competing interests:** The authors have declared that no competing interests exist.

to optimize their store's stock levels. The second reason is that they have incentive marketing to persuade clients to make multiple purchases in a single showroom or store. Remarkably, a management strategy for fashionable products can be improved by employing multiperiod inventory models and multi-objective optimization methods.

## 1.1 Relevant studies on heuristics and search technique

Recently, many metaheuristic algorithms have been presented for solving optimization problems mimicking and inspired by animal behavior and biological and natural phenomena. Due to the constantly ever-changing market environment, numerous studies on controlling inventory models have been presented. The holding costs and the necessary storage space for a multi-item multi-periodic inventory model were optimized by Mousavi et al. [1], which uses the Particle Swarm Optimization (PSO) technique. Also, it has been employed in a bi-objective inventory model for a two-echelon supply chain management. The PSO is one of the well-known optimization algorithms that belong to the metaheuristic algorithms [2]. Customer demand for seasonal goods exhibits a stochastic (uncertain) nature, which might be difficult for decision-makers. One of the critical components of a regular inventory management review is to ensure that it has an effective decision-support strategy when stochastic demands cause shortages [3]. Mohebbi et al. [4] presented multiple replenishment and multilayer delivery in conjunction with periodic evaluation. Numerous papers have discussed the multi-period inventory system. Taleizadeh et al. [5] used evolutionary algorithms to explore multiproduct, multi-constraint inventory management systems with stochastic replenishment intervals and discount rates. Then, the main issue was addressed using mixed integer nonlinear programming. Stochastic demand is also introduced to solve the inventory routing problem for fuel supply using two-stage programming [6]. The ideal or nearly ideal solution to inventory management issues could be viewed as the quarry of the hunting tactic. Metaheuristic algorithms based on the predatory behavior of herd animals have been employed to resolve various inventory management problems. Some researchers presented optimal solutions to their models using the PSO algorithm, for example, an inventory problem with deteriorating goods [7]. The solution to an integer nonlinear programming model with adjustable lead time was presented [8]. Metaheuristic algorithms are also used to solve multi-objective optimizations. Pasandideh et al. [9] developed a mixed binary integer mathematical program for multi-item, multi-period inventory ordering. They used Genetic Algorithms (GA) and Simulated Annealing (SA) to tackle the inventory problem. For the two-machine, no-wait flow shop, a more powerful heuristic is used to address the optimization of lateness [10]. Another approach that could be mentioned is the metaheuristic solution, which Turkeš et al. [11] applied to solve the stochastic facility location problem.

Recently, the following algorithms have attracted considerable attention from both academics and practitioners on heuristic optimization. The Quantum-based Avian Navigation Optimizer (QANO) was proposed by Zamani et al. [12], which utilizes quantum mechanics principles and bird navigation behavior to solve economic load dispatch problems. Nadimi-Shahraki et al. [13] introduced the Improved Moth-Flame Optimization (I-MFO) algorithm, which integrates the original Moth-Flame Optimization (MFO) algorithm and an adaptive mechanism. Nadimi-Shahraki et al. [14] developed the Diversity-Maintained Multi-Trial Vector Differential Evolution (DMDE) algorithm designed for large-scale global optimization problems with a Diversity Maintenance Strategy (DMS). Sahoo et al. [15] presented the Migration-based Moth-Flame Optimization (M-MFO) algorithm, which incorporates migration strategies into the MFO algorithm to enhance its search capabilities. These algorithms have been successfully applied to various optimization problems, such as economic load dispatch,

parameter tuning, and numerical optimization. Many studies have shown the effectiveness of these algorithms in solving complex optimization problems and their superiority over traditional optimization algorithms.

Metaheuristic optimization algorithms have become increasingly popular in solving complex problems due to their unique properties. Vasanthkumar et al. [16] introduced the Wild Horse Optimizer (WHO) for solving engineering problems, with applications to battery management systems for the Internet of Things (IoT) based hybrid electric vehicles. Minh et al. [17] proposed a new metaheuristic optimization based on the K-means clustering algorithm, which applies to structural damage identification. Kaveh et al. [18] improved Arithmetic Optimization Algorithm (AOA) to solve structural optimization problems. Ayyarao et al. [19] introduced the War Strategy Optimization (WSO) based on the strategic movement of army troops during the war, with an application to estimate the parameters of solar photovoltaic models. Finally, Zhang et al. [20] presented a modified African Vulture Optimization Algorithm (AVOA), which applies to the optimal model evaluation of the proton-exchange membrane fuel cells. These algorithms showcase diverse techniques for solving optimization problems, making them valuable tools in various fields.

The IoT with a collective network is a trending technology providing a new paradigm that merges the digital and physical universes, in which physical objects are embedded with sensors, software, and other technologies. Hosseini et al. [21] presented a novel method for detecting botnets in IoT by utilizing a feature selection technique based on the Slime Mold Algorithm (SMA), Salp Swarm Algorithm (SSA), and an efficient multi-objective algorithm. The proposed system can accurately identify botnets by selecting relevant features, demonstrating its effectiveness in the context of IoT. Sharma et al. [22] proposed the advanced butterfly optimization algorithm with a non-dominated sorting strategy to solve multi-objective problems. This algorithm offered a practical approach to solving complex optimization problems with multiple objectives. Also, data mining or machine learning problems were presented by [23]. They proposed a hybrid Multi-Objective Harris Hawks Optimization-based Fruitfly Optimization Algorithm (MOHHOFOA) to reduce the categories' processing time and accuracy. This survey article will be beneficial to scholars in finding relevant references in the field of scheduling, facility allocation, transportation, warehousing, and inventory management.

## 1.2 Comparison and evaluation of optimization algorithms

According to Safari et al. [24], MOGWO can be considered an efficient algorithm when compared to other algorithms such as Multi-Objective Water Cycle Algorithm (MOWCA), Multi-objective Particle Swarm Optimization (MOPSO), and Non-Dominated Sorting Genetic Algorithm-II (NSGA-II). The authors used a tri-objective mathematical model to solve the transportation-location-routing problem and compared the performance using four measures. The test results demonstrated that MOGWO was ranked first in realizing the tri-objective mathematical model. In addition, Heidari et al. [25] compared MOGWO and NSGA-II algorithms in green two-echelon closed and open location-routing problems. They found that the MOGWO algorithm generally performed better than the NSGA-II algorithm. These studies find that MOGWO is a new intelligent algorithm for solving multi-objective optimization problems, ensuring a better convergence rate than other optimization algorithms. When used in a practical case for inventory management, the GWO method might be more complicated, but it offers reliable performance in some circumstances. The robust grey wolf optimizer (RGWO) was used by Khalilpourazari and Khalilpourazary [26] to explore a mathematical model for maximizing production time in a multi-pass milling process. The multi-objective water cycle algorithm (MOWCA) and the multi-item EOQ model are included in the

application of MOGWO [27]. The paper aims to propose a more realistic inventory management system and multi-periodic inventory control model by gradually incorporating standard assumptions to address the usability issue. Demands are ultimately unknowable during each period due to a finite budget, and scarcity is also considered in the objective function. These realistic settings, which place several restrictions on the dynamic model, increase the applicability and complexity of the model. In essence, the dynamic model in this article is based on the work of Mirjalili et al. [28] and Mousavi et al. [1]. Still, there has been a modification to the models' characteristics. The demand is no longer constant but changes to be probabilistic under stochastic order size to produce a random demand.

The article is structured as follows. Section 2 introduces a random function to generate a stochastic demand that affects inventory dynamics. Decision variables are then employed to simulate the inventory operation, with the objective functions for profit and storage space. Section 3 focuses on developing and applying the intelligent MOGWO algorithm to provide an optimal solution for objective functions presented in Section 2. Section 4 implements the proposed algorithm for the inventory problems in Section 2, in which the numerical results and convergence process are described, along with a sensitivity analysis. Finally, Section 5 concludes the paper and discusses the study's limitations and future work.

## 2. Mathematical formulation of an inventory model

A company might randomly handle several items to meet customer needs over time within a finite planning horizon covering several periods. The planning horizon begins with a period and stops at a specific time. The shortage might impact the inventory carrying policy as the lost sale, and the storage space cost is formulated from the batch sizes of each item and the required space storage per unit item. The total available budget is limited and not changing. There exist limitations on ordering or production that regulate the order quantities of all items during different periods to their upper limits. Holding costs, backorder costs, and lost sales are associated with the inventory management policy. In addition, new production policies (such as installing a new manufacturing line, expanding a warehouse, or constructing a new storage area) will require optimum total storage space and profit. To optimize total profit and storage space simultaneously, it is necessary to identify the inventory levels of the items in each period.

### 2.1 Assumptions and notations

Before presenting the dynamic model, the assumptions are listed as follows.
Assumption:

I. An item's demand rate is independent of its competitors and changes over time.

II. It is possible to place only one order per period.

III. All items have a different initial inventory level at the beginning.

Terminology and notation involved in the dynamic model are multiple parameters. The definition and notation of these parameters can be found in Table 1.

### 2.2 Problem formulation

Figs 1 and 2 describe inventory management through the changes in stochastic demand. The initial inventory begins with a random order quantity at the primary period ($T_0$). The inventory level decreases when demand occurs at any unit of time. If the demand still occurs, the inventory level continuously drops until it reaches zero. If a new order is not placed to meet the demand, a shortage occurs, leading to lost sale costs for all items that could not be sold in

**Table 1. Notations of inventory parameters and variables.**

| | |
|---|---|
| **Indices** | |
| $i = 1, 2, \ldots, I; j = 1, 2, \ldots, J$ | |
| **Parameters** | |
| $I$ | Number of periods during the planning horizon |
| $J$ | Number of items or products |
| $RS_i$ | Required storage space per unit of the $i^{\text{th}}$ item |
| $T_j$ | Total time duration including the $j^{\text{th}}$ period |
| $t'_{i,j}$ | $j^{\text{th}}$ period in which the inventory level of item $i$ is zero |
| $SP_i$ | Price of item $i$ |
| $Sr_{i,j}$ | Total number of quantity sold item $i$ in period $j$ |
| $PQ_{i,j}$ | Purchase quantity of item $i$ in period $j$ |
| $OC_i$ | Ordering cost per period of item $i$ |
| $IP_{i,j}$ | Inventory position of the $i$th item in period $j$ |
| $IL_i$ $(t)$ | Inventory level of the $i^{\text{th}}$ item at time $t$ |
| $UH_i$ | Unit inventory holding cost for item $i$ |
| $PC_i$ | Purchasing cost per unit of item $i$ |
| $Csc_{i,j}$ | Shortage cost per unit of the $i$ product in period $j$ |
| $BS_i$ | Batch size of item $i$ |
| $U_{i,j}$ | Number of packets |
| $\lambda$ | Weight of profit function |
| $\mu$ | Weight of store space function |
| $R$ | Total sale price |
| $O$ | Ordering cost |
| $H$ | Holding cost |
| $L$ | Lost sale cost |
| $TB$ | Total budget |
| $\Omega_\Sigma$ | Total profit |
| $\Pi_\Sigma$ | Total storage space |
| $D$ | Range from the prey to the grey wolf |
| $X$ | Position vector of the grey wolves |
| $A, C$ | Vectors of coefficients |
| $TFF$ | Total fitness function |
| **Stochastic parameters** | |
| $D_{i,j}$ | Demand of the $i^{\text{th}}$ item in period $j$ |
| $Po_{i,j}$ | A float variable ($\in [0,1]$) representing the probability that item $i$ will be bought in period $j$ |
| $So_{i,j}$ | An integer variable describing the size or the quantity of item $i$ when it is ordered in period $j$ |
| **Decision variables** | |
| $Q_{i,j}$ | Order quantity of item $i$ at the end of period $j$ or the beginning positive inventory level of the $i$ item in period $j$ |
| $r_{i,j}$ | Reorder point of item $i$ in period $j$ |

the inventory. The time when a shortage occurs till a new order is made will be denoted as $t$, while a period that includes the shortage time is $T$ ($t < T$).

Consumer demand for each product is continuously changing as time goes on. Therefore, the change in demand can be described using a stochastic process. Providing an exact estimate of a customer's probability of ordering on any given day is possibly tricky. Still, their chance of placing a purchase on any given day with a probability $Po_{i,j}$ could be estimated. This stochastic

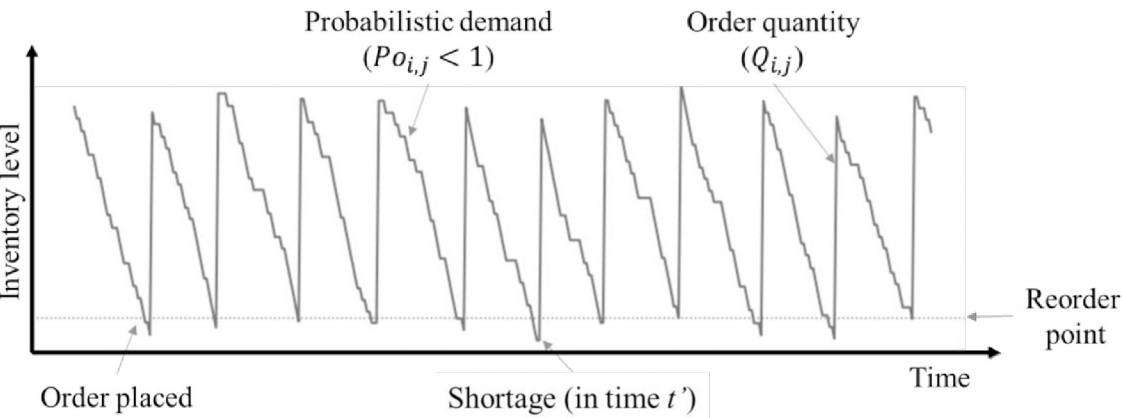

**Fig 1. Inventory level of an item with multiple periods (discrete demand).**

(or random) variable is also created by using a uniform distribution, which has the probability density function given as follows:

$$Po_{i,j} = \frac{1}{b - a} \tag{1}$$

where $a = 0$ is the lower limit and $b < 1$ is the upper limit with the interval $[a, b)$. The order size is another uncertainty without a valid contract with a particular client. According to the assumptions for this study, the order size has a log-normal distribution whose parameters are unknown. A continuous probability distribution of a random variable is known in probability theory as a log-normal (or log-normal) distribution. As a result, $Y = \ln(X)$ has a normal distribution if the random variable $X$ is a log-normal distribution. The exponential function of $Y$, or $X = \exp(Y)$, has a log-normal distribution if $Y$ has a normal distribution. A random variable that is log-normally distributed takes only positive fundamental values. It will be a practical model for measurements in engineering and exact sciences, medicine, economics, and other fields (such as energies, concentrations, lengths, and financial prices). If the natural logarithm of $X$ is typically distributed, then $X$ is log-normally distributed with mean $\mu$ and variance $\sigma^2$:

$$\ln(X) \sim N(\mu, \sigma^2) \tag{2}$$

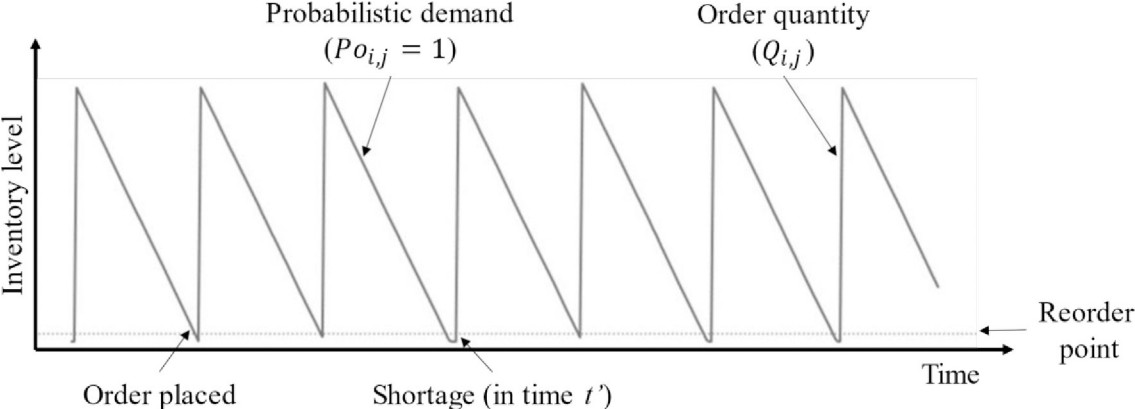

**Fig 2. Inventory level of an item with multiple periods (continuous demand).**

Then, the log-normal distribution could be applied to demonstrate the other size as follows:

$$f(t) = \frac{1}{t\sigma\sqrt{2\pi}} \exp\left(-\frac{(\ln t - \mu)^2}{2\sigma^2}\right) \tag{3}$$

Therefore, the demand function presents the order of item $i$, which has $Po_{i,j}$ percentage order at time $t$:

$$D_{i,j} = \begin{cases} 0 & , Po_{i,j} < P \\ \dfrac{1}{t\sigma\sqrt{2\pi}} \exp\left(-\dfrac{(\ln t - \mu)^2}{2\sigma^2}\right) & , Po_{i,j} \geq P \end{cases} \tag{4}$$

With the above formulation, the total profit is now given below

$$\Omega_\Sigma = R - (O + H + L) \tag{5}$$

where $O$ denotes total ordering cost; $H$ describes total holding cost; $L$ is a total shortage or lost sale cost; $R$ means total selling price. The total selling price is given by,

$$R = \sum_{i=1}^{I}\sum_{j=1}^{J} SP_i \times Sr_{i,j} \tag{6}$$

Whenever an order is placed for an item in a period, an ordering cost ($O$) exists. Orders can be placed after different lead times when the inventory level drops below the reorder point. As a result, the total ordering cost is calculated as follows:

$$O = \sum_{i=1}^{I}\sum_{j=1}^{J-1} OC_i \times D_{i,j} \tag{7}$$

The holding (carrying) cost calculation requires consideration of the possibility of shortages for a given product in a period, which is different from the ordering cost. When the shortage occurs in period $j$, a binary variable $Sh_{i,j}$ is applied to calculate the holding cost. If $Sh_{i,j} = 1$, a shortage occurs in period $j$ and 0 otherwise. For item $i$ in the time interval $T_{j-1} \leq t \leq T_j(1 - Sh_{i,j}) + t\prime_{i,j}Sh_{i,j}$, its holding cost is described below,

$$H_i = \int_{T_{j-1}}^{T_j(1-Sh_{i,j})+t'_{i,j}Sh_{i,j}} Q_i(t)dt \tag{8}$$

In this formulation, $Sh_{i,j}$ equals 1 in case of shortage of item $i$, and the term $T_j(1 - Sh_{i,j}) + t\prime_{i,j}Sh_{i,j}$ turns into $t'_{i,j}$; otherwise, $Sh_{i,j} = 0$ and $T_j(1 - Sh_{i,j}) + t\prime_{i,j}Sh_{i,j} = 0$. The inventory position of the item in that period is calculated by subtracting the demand of item $i$ at period $j$ from the inventory level, and the order quantity is given by

$$IL_{i,j+1} = IL_{i,j} + So_{i,j} - D_{i,j} \tag{9}$$

Finally, the total holding cost in a warehouse is obtained as follows:

$$H = \begin{cases} \sum_{i=1}^{I}\sum_{j=1}^{J-1} \left( \dfrac{IL_{i,j} + So_{i,j} - D_{i,j}}{2} \right) \times T_j & , Sh_{i,j} = 0 \\ \sum_{i=1}^{I}\sum_{j=1}^{J-1} \left( \dfrac{IL_{i,j} + So_{i,j} - D_{i,j}}{2} \right) \times t\prime_j & , Sh_{i,j} = 1 \end{cases} \tag{10}$$

The total shortage cost includes the lost sale cost. As shown in Fig 1, the trapezoidal area in each period, multiplied by the lost sale cost per unit demand of the $i$ product in period $j$, yield $L$, which becomes the lost sale cost of the item in that period. Then, the total lost sale cost will be given by,

$$L = \sum_{i=1}^{I}\sum_{j=1}^{J-1} \left( \frac{Qsc_{i,j} \times Csc_{i,j}}{2} \times \left( T_j - t\prime_{i,j} \right) \right) \tag{11}$$

Hence, the first objective function of the optimization problem can be expressed by,

$$\Omega_{\Sigma} = \sum_{i=1}^{I}\sum_{j=1}^{J} SP_i \times Sr_{i,j} - \left( \begin{array}{l} \sum_{i=1}^{I}\sum_{j=1}^{J-1} OC_i \times D_{i,j} \\ +\sum_{i=1}^{I}\sum_{j=1}^{J-1} \left( \dfrac{IL_{i,j} + So_{i,j} - D_{i,j}}{2} \right) \times \left( T_j(1 - Sh_{i,j}) + t'_{i,j} Sh_{i,j} \right) \\ +\sum_{i=1}^{I}\sum_{j=1}^{J-1} \left( \dfrac{Qsc_{i,j} \times Osc_{i,j}}{2} \times \left( T_j - t'_{i,j} \right) \times Up_{i,j} \right) \end{array} \right) \tag{12}$$

The second target for the inventory problem is optimizing the required storage space. The inventory level at the beginning of a period is equal to the order quantity ($Q$) at this period plus with inventory level of the previous period. Since an amount of order arrives to replenish the storage at each period, the objective function $\Pi_{\Sigma}$ is described by,

$$\Pi_{\Sigma} = \sum_{i=1}^{I}\sum_{j=1}^{J-1} (Q_{i,j} + PQ_{i,j}) RS_i \tag{13}$$

Since $PQ_{i,j}$ represents the purchased quantity of item $i$ in period $j$, by denoting the batch size as $B_i$ and the number of packets as $U_{i,j}$, then it becomes,

$$PQ_{i,j} = BS_i U_{i,j} \tag{14}$$

The total fitness function (*TFF*) is the combination of maximizing profit or max ($\Omega_{\Sigma}$) and minimizing storage space cost or min ($\Pi_{\Sigma}$), where $\lambda$ and $\mu$ are the weights in the total fitness function, typically $0 \leq \lambda \leq 1$ and $0 \leq \mu \leq 1$, ensuring $\lambda + \mu = 1$. To simplify the *TFF*, maximizing profit could be converted to the minimum problem $\lambda \Phi_{\Sigma} = -\lambda \Omega_{\Sigma}$. Then, *TFF* is described by the minimum value of a sum of two values below:

$$MinTFF = \lambda \Phi_{\Sigma} + \mu \Pi_{\Sigma} \tag{15}$$

subject to

$$IL_{i,j+1} = IL_{i,j} + So_{i,j} - D_{i,j};$$

$$PQ_{i,j} = BS_i U_{i,j}; \sum_{i=1}^{I} \sum_{j=1}^{J-1} PQ_{i,j} PC_i \leq TB; PQ_{i,j} \leq M_1; PQ_{i,j+1} \geq Osc_{i,j} \qquad (16)$$

Referring to Eqs (15) and (16), the objective function describes the quantity that needs optimization or the function to be minimized or maximized. The objective function is subject to linear equality and inequality constraints on the decision variables. The constraints are restrictions set on the primary objective function that needs to be satisfied. A feasible solution is optimal if its objective function value equals the smallest value *TFF* can take over the possible space.

Optimizing the inventory model balances asset constraints against multiple objectives and fulfills business goals across inventory stock units. So far, most studies have imposed unrealistic assumptions on inventory-planning models. Obtaining an acceptable solution for integer nonlinear optimization problems under practical constraints will be challenging for most researchers. If an order of *Q* units arrives each time, there is a jump in inventory level, and the inventory decreases at a non-static rate based on probabilistic demand. A new order might be placed once the inventory level drops to reorder point. At some point, higher demand will cause a stock-out before a new order is received. Next, the efficient metaheuristic algorithm is presented for solving inventory optimization problems.

## 2.3 Inventory model simulation by Monte Carlo method

The Monte Carlo experiment is a commonly utilized computational algorithm using a probabilistic numerical technique that predicts possible outcomes of an uncertain event. This technique typically involves conducting numerical experiments to acquire statistics for output variables in a computational model, considering sample statistics of input variables. In each experiment, random input variables are sampled according to their distributions, and the computational model is used to calculate output variables. Multiple scenarios can be generated to obtain statistics for the output variables of a computational inventory model using the probability distributions of input variables. This method is required to determine the probability distribution of demand for each item in the inventory, which can be achieved by analyzing historical data, market trends, and other relevant factors. The Monte Carlo simulation generates scenarios by randomly sampling the demand for each item from its probability distribution. Subsequently, the inventory levels for each item are calculated based on the demand values generated. The inventory's behavior can be distributed by running the Monte Carlo method multiple times. The evidence gathered from this analysis can offer information to help policymakers make informed decisions on inventory management, such as setting safety stock levels, reorder points, and order quantities. The pseudocode for applying Monte Carlo simulation to an inventory model is provided in Table 2.

## 3. Multi-objective optimization using the GWO algorithm

The grey wolf algorithm mimicking wolves' social structures and hunting behavior in multi-objective search spaces is a novel swarm intelligence and metaheuristic algorithm [28]. First, it is essential to note that grey wolves hunt in groups and have a hierarchy within their herds. Wolves will take on quests of the corresponding difficulty with higher ranks, and levels will also be adjusted to match related requirements as they decrease. They have a strict social dominance hierarchy, as depicted in Fig 3.

**Table 2. Pseudocode of Monte Carlo in inventory simulation.**

| Method: Inventory simulation by Monte Carlo method | |
| --- | --- |
| Initialize inventory levels for each item $IL_i(0)$ | |
| Set the number of Monte Carlo simulations to be conducted ($N_{Monte}$) | |
| For each Monte Carlo simulation | |
| | Sample the demand ($D_i$) for each item from its corresponding probability distribution |
| | Calculate the inventory levels $IL_i(t)$ for each item based on the demand values generated ($D_i$) |
| | Record the final inventory levels for each item $IL_i(T)$ |
| End the *For-loop* simulation | |
| Return all the inventory levels during the period $T$ | |

## 3.1 The action of surrounding the prey

The first step in hunting, when the wolves detect the prey, is encircling it. The mathematical model for surrounding actions is given as follows:

$$D = |C \times X_i(k) - X(k)| \tag{17}$$

$$X_i(k+1) = X(k) - A \times D \tag{18}$$

where $A$ and $C$ are the vectors of coefficients, $k$ is the iteration number, $X(k)$ is the position vector of the grey wolves, $X_i(k)$ is the goal position vector or the prey position vector of the grey wolves, and $D$ is the range from the prey to the grey wolf. Also, the coefficient vectors are described below,

$$A = 2 \times h \times l_1 - h \tag{19}$$

$$C = 2 \times l_2 \tag{20}$$

$$h = 2 - i \times \left(\frac{2}{k_{\max}}\right) \tag{21}$$

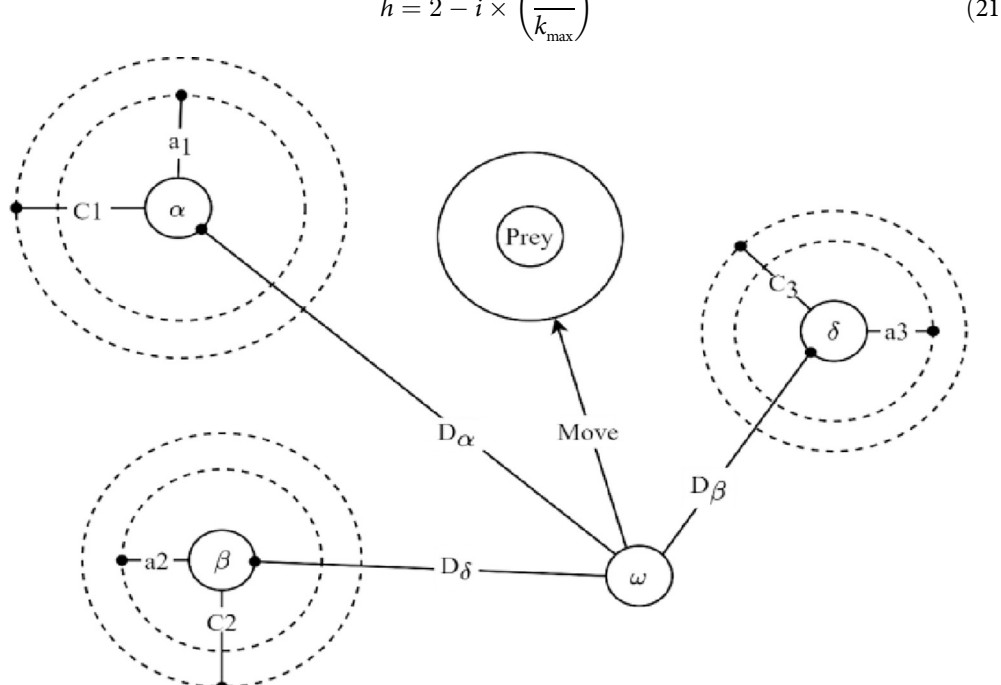

**Fig 3. Implementation of positions updating in grey wolf optimizer.**

where the parameter $h$ is linearly decreased from 2 to 0 throughout iterations; $l_1$, and $l_2$ are random vectors in [0, 1].

## 3.2 Hunting mechanism

Due to the robust prey detection ability, the social behavior of grey wolves is one of the highest hunting success rates in the wildlife world. Alpha ($\alpha$), beta ($\beta$), and delta ($\delta$) are described as the leader and the best alternatives for alpha. In the algorithm, the alpha is depicted as the optimal solution, while beta and delta could provide prey knowledge to alpha. Assume that the alpha, beta, and delta have a greater understanding of the prospective locations of prey to imitate the hunting activities. Accordingly, alpha, beta, and delta are the first three optimal solutions, requiring the other search agents to replace their locations. Their updated locations are described as follows:

$$D_\alpha = |C_1 \times X_\alpha(k) - X(k)|; D_\beta = |C_2 \times X_\beta(k) - X(k)|; D_\alpha = |C_3 \times X_\delta(k) - X(k)| \qquad (22)$$

$$X_1 = X_\alpha - A_1 \times D_\alpha; X_2 = X_\beta - A_2 \times D_\beta; X_3 = X_\delta - A_3 \times D_\delta \qquad (23)$$

$$X(k+1) = \frac{X_1 + X_2 + X_3}{3} \qquad (24)$$

where $X_\alpha$, $X_\beta$, and $X_\delta$ are the present locations of $\alpha$, $\beta$, and $\delta$, respectively; $X(k)$ is the vector of the prey locations of the grey wolves; $D_\alpha$, $D_\beta$, and $D_\delta$ are the ranges from the prey to $\alpha$, $\beta$, and $\delta$, respectively; $X(k+1)$ describe the location vector with updating searching factor while $A$ and $C$ are the random vectors. Fig 3 illustrates the operation of a search agent that updates its location by following alpha, beta, and delta. From the indication, the final position could be a random point within the range surrounded by alpha, beta, and delta. Three optimal wolves will forecast the following location of the prey and randomly update their new locations around the prey. The search agent will continue until the distance from the prey to the wolves is minimal.

Multi-objective optimization using GWO is realized by integrating two new components. The Pareto archived evolution strategy stores non-dominated Pareto optimum solutions found so far. In addition, the leader selection strategies from the alpha, beta, and delta solutions can be used for selecting the leaders of the hunting process. A storage unit has been developed to save or retrieve non-dominated Pareto solutions. Members archived are limited to a certain number, and a comparison is conducted between non-dominated solutions obtained and achieved residents during iteration. Due to the limitations of the archived population, some restrictions are described as follows:

- The new solution dominates at least one solution in the archive, and the updated solution is allowed into the archive if the dominated solutions are removed.

- The newly dominated solutions should be added to archives if neither they nor the archive members outperform each other.

- The grid mechanism should be run first if the archive is full of omitting one of the solutions in the most crowded segment. The solution should be inserted into the least crowded part to ensure that the final approximated Pareto optimal front has greater diversity.

First, the most crowded segments are selected when solutions are removed from a complete archive. One solution is randomly removed from one of them to accommodate the new solution. The second component is the leader selection mechanism. Three of the most effective

solutions obtained so far are used as alpha, beta, and delta wolves. The other search agents are directed toward promising regions of the search space to find a solution close to the global optimum. However, the Pareto optimality concept makes comparing solutions in a multi-objective search space difficult. The leader selection mechanism chooses the minor crowded portion of the searching space and provides one of its non-outperforming solutions as alpha, beta, or delta wolves. A roulette-wheel selection finishes the alternatives option with the following probability for each hypercube problem:

$$P_i = \frac{c}{V_i} \tag{25}$$

where $c$ is a constant greater than one, and $V$ is the number of the obtained Pareto optimal solutions in the $i^{th}$ segment. The pseudocode of the MOGWO method is described in Table 3, and an illustrative overview of the proposed algorithm is presented in Fig 4.

In this algorithm, the optimum solutions for real-world problems pursue multiple goals. Several observations are made on how the MOGWO could be effective in providing the optimal solutions to multi-objective problems:

**Table 3. Pseudocode of multi-objective optimization.**

| **Algorithm: Multi-objective grey wolf optimizer (MOGWO)** | | |
|---|---|---|
| Declare the population of the grey wolf | | |
| Initialize the $a$, $A$, $C$, maximum number of iterations ($l_{max}$), and search agents ($n$) | | |
| Calculate the objective values for each search agent | | |
| **For** $i$ = 1: $n$ do | | |
| | Calculate the total fitness function (TFF) | |
| **End for** | | |
| Find the optimal solution of each search agent ($X_\alpha$, $X_\beta$, and $X_\delta$) | | |
| Set iteration counter $l$ = 1 | | |
| **For** $l$ = 1: *Max number of iterations* do | | |
| | Update $A$ and $C$ by Eq (19) and Eq (20) and each search agent; Decrease a from 2 to 0; | |
| | Calculate the fitness function (solutions) | |
| | Update the solutions ($X_\alpha$, $X_\beta$, and $X_\delta$) by Eq (23) | |
| | Calculate the fitness (solutions) for each search agent | |
| | Select the non-dominated optimal solutions | |
| | Update the archive based on the obtained non-dominated optimal solutions | |
| | **If** the archive is filled | |
| | | Remove a few solutions using the roulette wheel from the archive |
| | **End if** | |
| | **If** any fresh solutions to the archive are positioned outside the hypercubes | |
| | | Grid is updated to protect the novel solutions |
| | **End if** | |
| | $X_\alpha$ = Select leader (archive) | |
| | Exclude $\alpha$ temporarily from the archive to evade picking the same leader | |
| | $X_\beta$ = Select leader (archive) | |
| | Exclude $\beta$ temporarily from the archive to evade selecting the same leader | |
| | $X_\delta$ = Select leader (archive). Add $\alpha$ and $\beta$ again with the archive | |
| | $l$ = $l$ + 1; | |
| **End for** | | |
| Return the archive | | |

- An external archive saves the best non-outperform solutions obtained so far.

- The search agents are permitted to locate the expected position of the prey due to MOGWO becoming heir to the hunting mechanism.

- Exploration and exploitation are guaranteed by the adaptive values of $a$ and $A$.

- Non-adaptive random values for the $C$ parameter during optimization enhance exploration and the local front immunity of the MOGWO algorithm concomitantly.

Based on these observations, the weighted sum method, which uses weight factors between 0 and 1, is a strategy that may be broadly applied to different situations with many objectives. To determine how essential each objective function is to the overarching goals, they are multiplied by each objective value. This strategy might be helpful when two or more objective functions share the same unit or have objective values within the same range. In different scales, a considerable objective value may completely outweigh a small one, even though the smaller value has a greater weight. With new mechanisms such as an archive function and a domination detector function, MOGWO could perform bi-objective functions together to find optimized value without employing any weight. While the profit (USD) and storage space ($m^2$) have complete dissimilarity, the range of values has a big difference.

After achieving the demand for each item from Eq (4), the Monte Carlo method also simulates the inventory behavior. Monte Carlo technique applies Eqs (12) and (13) in Section 2 to find the objective function values. This process can be used to calculate fitness values for each search agent, as mentioned in Fig 4.

## 4. Numerical simulation

### 4.1 Stochastic demand generation

Naturally, inventory management problems pursue multiple objective functions which are generally conflicting. Several software solutions on the market aim to help decision-makers solve inventory optimization problems. The solution framework employs nonlinear programming for an optimization problem in this study. Four scenarios with various sizes and populations are utilized to verify the efficacy of the optimal solutions implemented by the MOGWO algorithm. More specifically, the proposed approach is a population-based metaheuristic algorithm that optimizes an inventory management problem by having a population of candidate solutions and moving the particles around in the search space. As shown in Table 4, the mean and standard deviation of each item's demand is provided to generate random demand by following log-normal distribution. The order parameters and the initial stock (ordering, holding, and storage space costs) are provided.

Fig 5 demonstrates the demand profiles for the first item, and the following graphs illustrate how another demand changed over a year for the remaining items. The illustration's demand for items 1 and 2 has never been at level zero since their probability is 1.0 (Table 4). Items 3 and 4 have a probability of 0.8 and 0.6, respectively, meaning that random demand may fall to zero occasionally. In detail, Item 1's probability is 1.0, which means this item is always bought on any day in 365 days. The quantity purchased daily is also completely different, created based on a log-normal distribution, with mean and standard deviation in Table 4. Item 2's demand pattern is calculated and simulated as the same method as item 1. With item 3, its probability is 0.8, which means, on any day in the periods, there is just 80% that item 3 is bought, and the purchased quantity on that day is also based on the log-normal distribution (mean and standard deviation needed to create distribution are also from Table 4 and similarly with item 4.

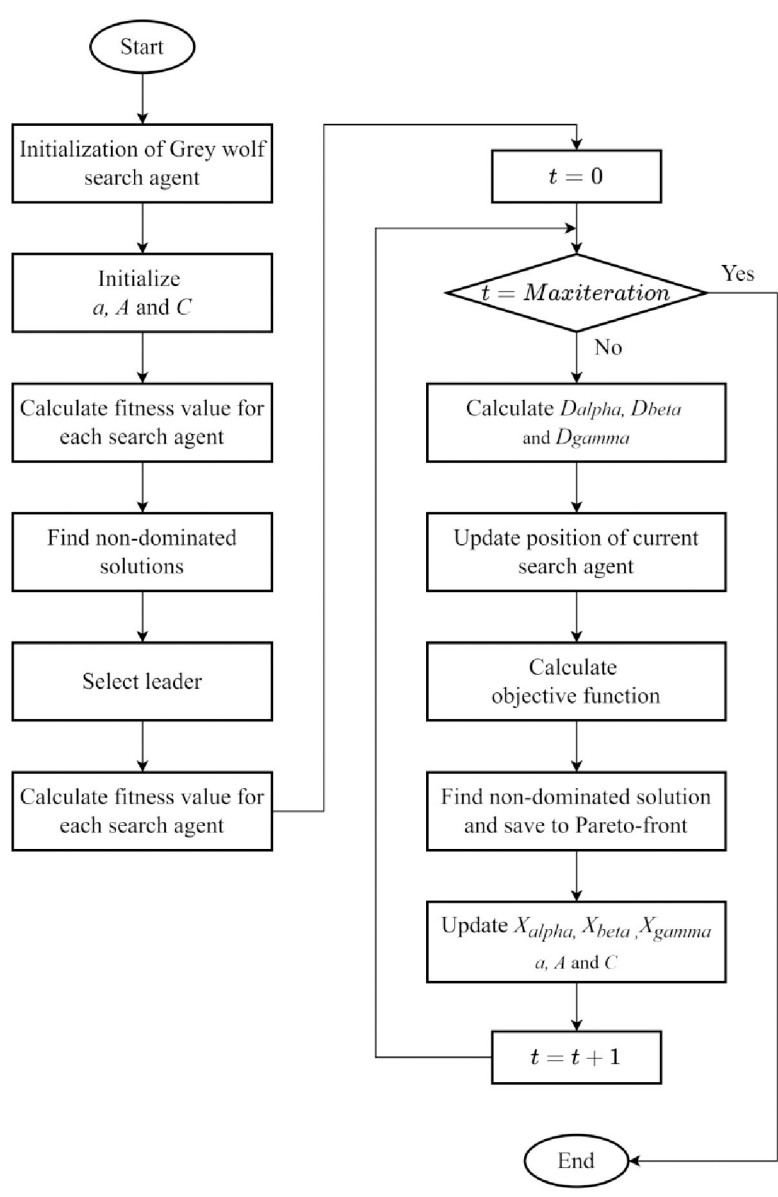

**Fig 4. Flowchart of the basic MOGWO algorithm.**

## 4.2 Convergence trend of a metaheuristic algorithm

This study demonstrates multiple objectives for optimizing the inventory management model, which deals with optimizing the profit and storage areas. The convergences of the optimal

**Table 4. Initial parameters of inventory items.**

| Item | Mean demand | Std dv demand | Starting stock | Selling price | Ordering cost | Holding cost | Space cost | Probability |
|------|-------------|---------------|----------------|---------------|---------------|--------------|------------|-------------|
| 1 | 5714 | 150 | 5813 | 50 | 1309 | 28 | 5 | 1.0 |
| 2 | 4199 | 245 | 4292 | 43 | 1235 | 32 | 6 | 1.0 |
| 3 | 4674 | 295 | 4860 | 15 | 811 | 37 | 6 | 0.8 |
| 4 | 5131 | 195 | 5128 | 50 | 933 | 34 | 4 | 0.6 |

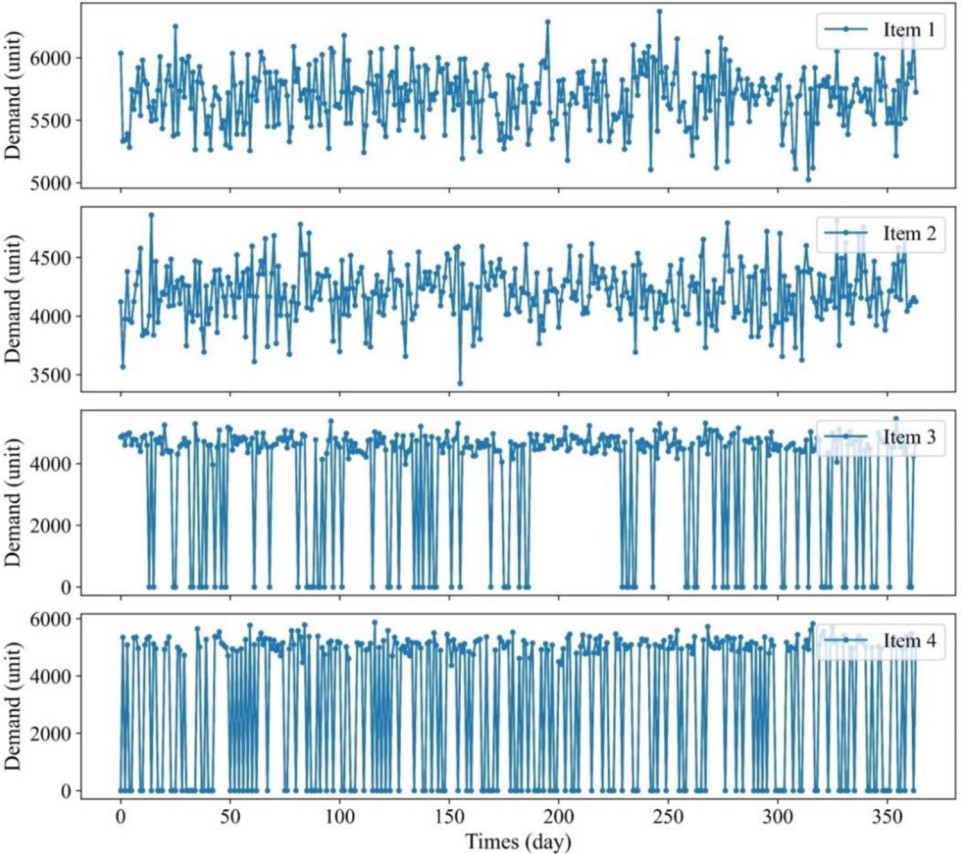

**Fig 5. Random demand for four items in 365 days.**

solutions are illustrated in different items to demonstrate how the optimal values are recorded. For each item, the convergence occurs differently with different effects of random demands. Therefore, the MOGWO algorithm proves that after a certain number of iterations, the scattered particles representing the candidate solutions will converge to a point or a specific zone, which is a near-optimal value and satisfies the objective functions mentioned before. The convergence trends of iterative methods for all the solutions are presented in Figs 6–9, illustrating the searching process of optimal solution space for multiple objectives in detail. The objective function on storage space is displayed by the vertical axis ($m^2$), while the objective function on profit is described by the horizontal axis (USD). For a graph, the convergence process of each item is illustrated by six sub-diagrams. Each sub-diagram shows a convergence stage if the convergence occurs in 50 iterations. For example, the first iteration is demonstrated by the first sub-diagram (left-top), and the five sub-diagrams show the convergence stages at the 10th, 20th, 30th, 40th, and 50th.

Item 1 starts at the first iteration, and 100 solutions are distributed in search space (Fig 6). The horizontal axis displays the profit (USD) to be maximized in each diagram. In contrast, the vertical axis presents the space unit ($m^2$) to be minimized. One hundred solutions are randomly generated based on specified parameters in the algorithm. At this stage, a set of solutions does not indicate the optimum, like a pack of wolves starting to search for prey. After 10, 20, 30, and 40 iterations, the solutions gradually converge to a specific region as wolves finally catch their prey, as demonstrated in the 50th iteration.

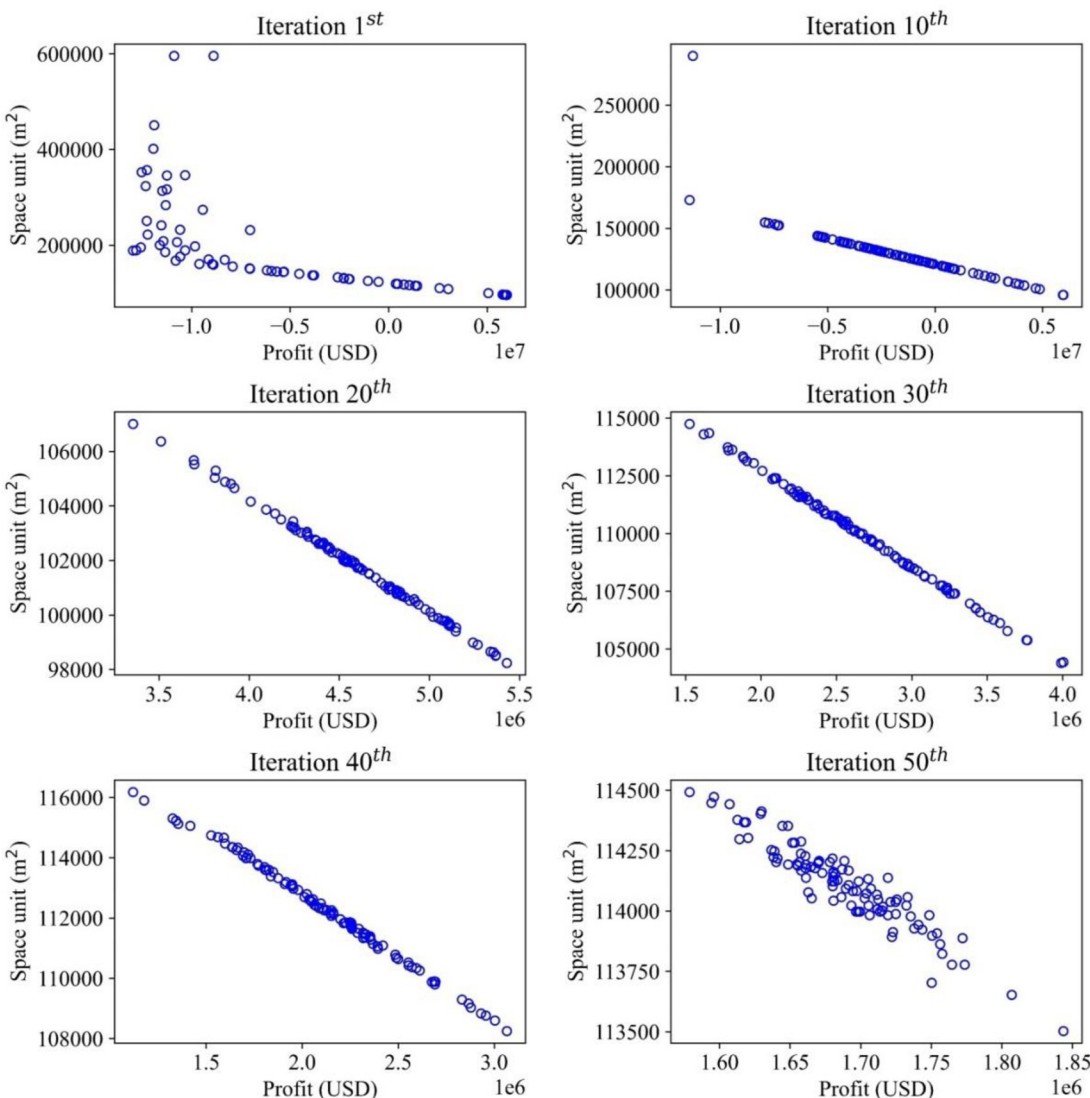

**Fig 6. Convergence analysis for the optimal solutions in the search space (Item 1).**

At the first iteration, the maximum value of storage space is 600000 (m²), and the minimum profit value is about -10000000 (USD). Then, at the final iteration, the maximum storage space value is 114500 (m²), and the minimized profit is 1600000 (USD). In contrast to the other items' solutions, the solution space tends to cluster only on a narrow range of values in the final iteration, making it relatively hard to identify an optimal solution. The convergence process directly yields the optimal solution that maximizes profit and minimizes storage space simultaneously.

In the case of item 2, the convergence trends in realizing multiple objectives are also displayed in Fig 7. Their behaviors are demonstrated as follows: from the first iteration, all solutions are distributed from -4000000 to 6000000 (USD) for maximizing profit and a range of 100000–600000 (m²) for minimizing storage space. After the 10th, 20th, 30th, and 40th iterations, the solutions find the optimal position in the search space. At the final iteration, two objective functions have achieved reliable results, ensuring profit ranging from 6100000 to 6180000 (USD) while the storage space ranges from 4300 to 4600 (m²). In the final phase of

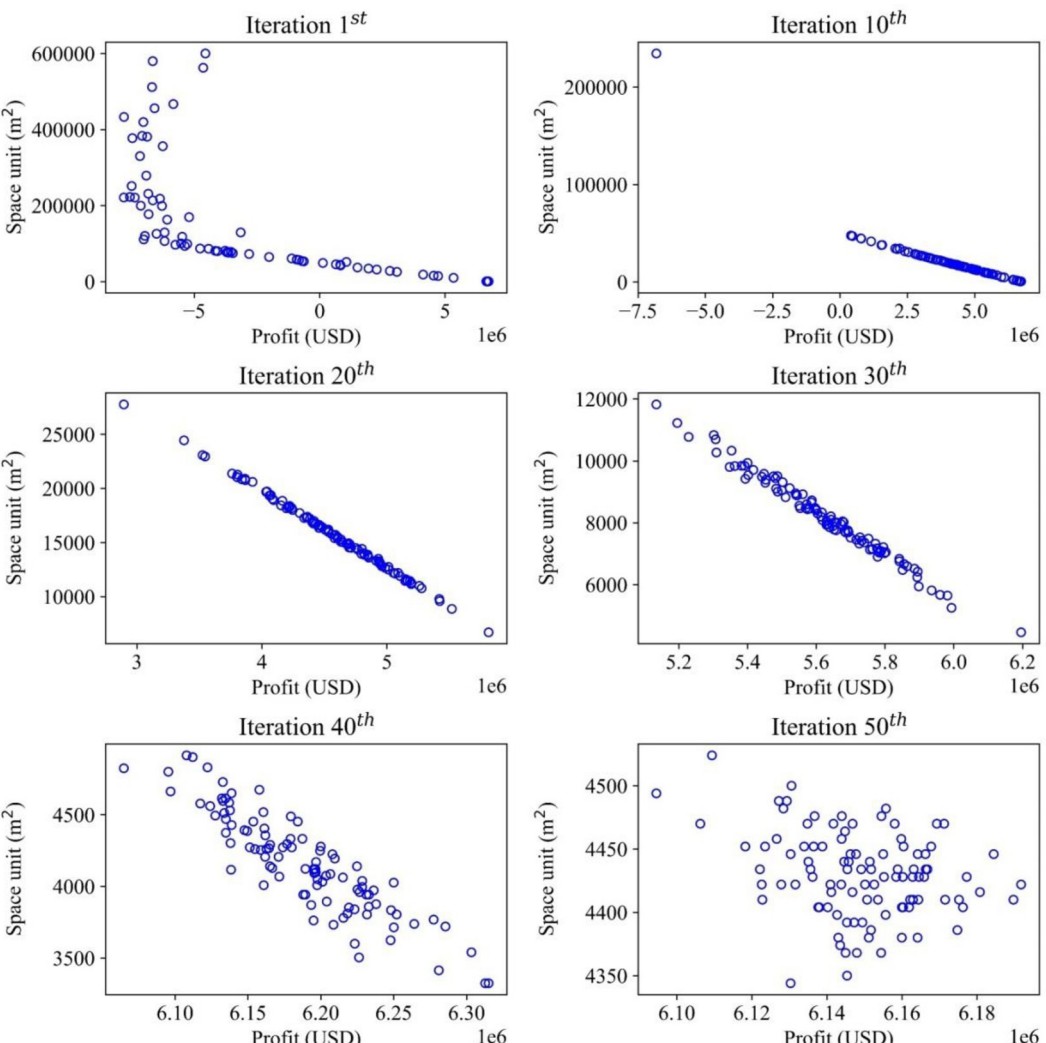

**Fig 7. Convergence analysis for the optimal solutions in search space (Item 2).**

the optimization process, it is observed that the objective function values are converged to much smaller ranges, as previously discussed. In contrast to the case of item 1, none of the solutions at the 50th iteration (final stage) completely dominates each other. Consequently, it may be appropriate to apply weighted sum methods at this stage to determine the relative importance of each objective function. In a multi-objective optimization problem, a solution's goodness is practically determined by dominance.

Fig 8 illustrates the optimization process of item 3, which is also implemented with the proposed algorithm to explore the optimal solutions. In the first iteration, the particles mainly remain in ranges of -4000000–5000000 (USD) for profit and 100000–600000 (m$^2$) for storage space. From the 30$^{th}$ iteration, the distribution changes significantly, and the values fall within ranges of -2000000 to 2000000 (USD) and 30000 to 60000 (m$^2$). At the 100$^{th}$ iteration, those particles converge to 1200000–2200000 (USD) and 22400–23200 (m$^2$). The optimization pattern of item 3 exhibits similar behavior to item 2. The final convergence stage indicates the convergence of all the objective values to a smaller range than the initial stages. Furthermore, the solutions that satisfy the dual requirements of maximizing profit while minimizing space

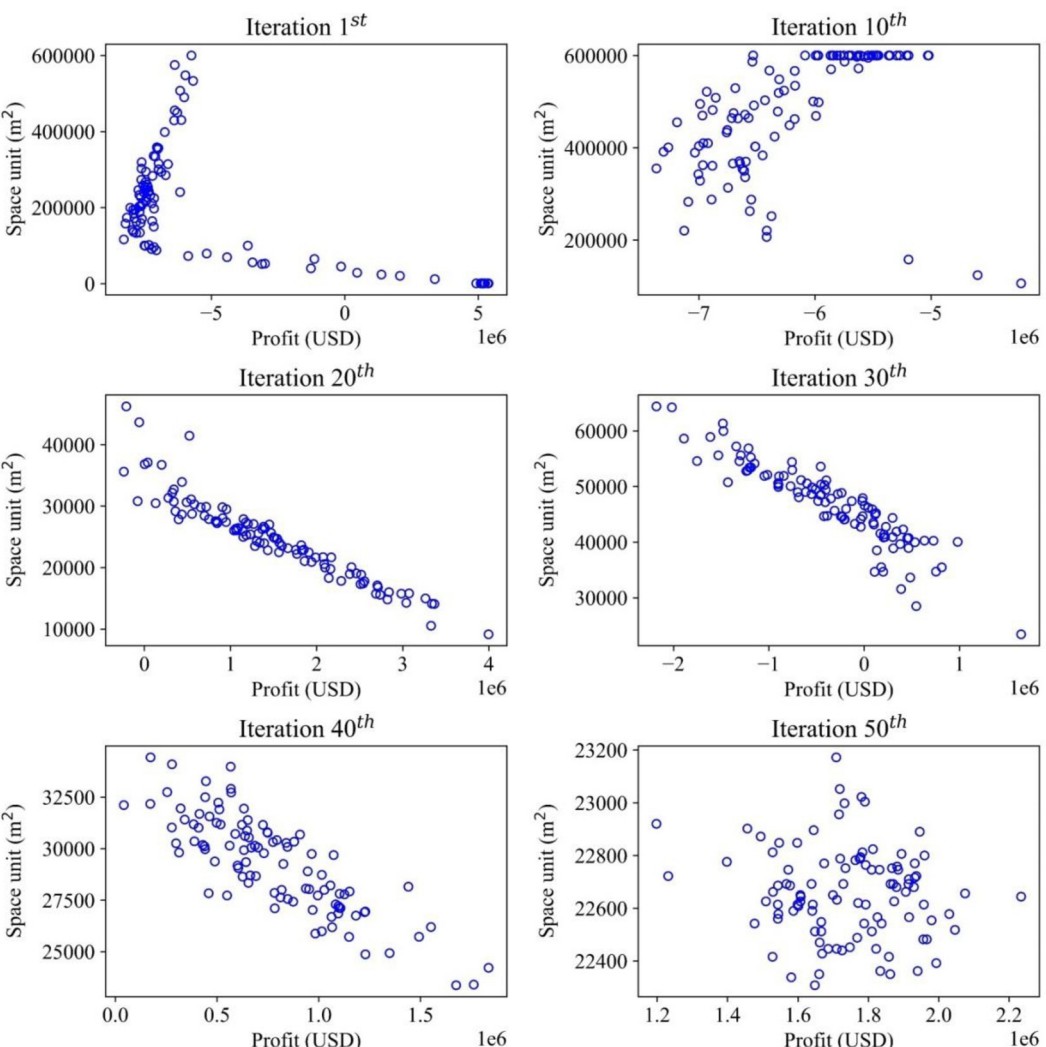

**Fig 8. Convergence analysis for optimal solutions in the search space (Item 3).**

are identified. At this stage, conclusively identifying the superior solution will be challenging without utilizing weight-based methods.

For item 4$^{th}$, the convergence process is shown by six diagrams in Fig 9, which also have the same expression as other items' optimization processes. The first iteration has an extensive range of values, 7500000–5000000 (USD) for the profit and 100000–300000 (m$^2$) for the storage space. The specified range of objective functions has decreased considerably after just ten iterations. For the multiple goals, they drop to the range of values within 3000000–5000000 (USD) and a new range of 1000–4500 (m$^2$). Through each iteration, multiple goals have achieved new optimal ranges within 4200000–5200000 (USD) for profit goal and 580–650 (m$^2$) for storage space. As a result, a search space size drastically reduces the computational cost to obtain an optimal solution.

As illustrated in Figs 5–8, the convergence processes of four items are all guaranteed. In the first iterations of each item, all the particles are placed randomly in different positions of the solution spaces, which could not reveal optimal points for objective functions. After 30 iterations, all the particles tend to move together to a specific region. Finally, all the particles are

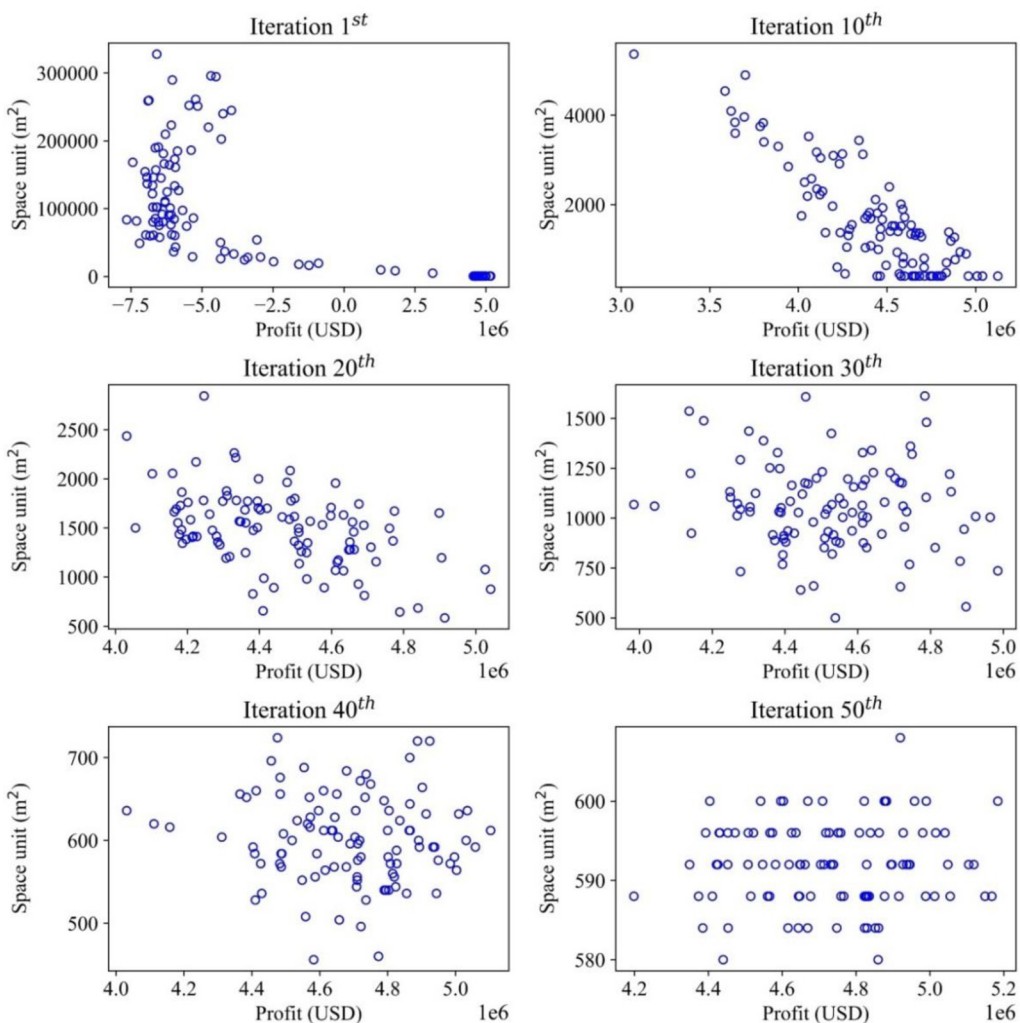

**Fig 9. Convergence analysis for optimal solutions in the search space (Item 4).**

guided toward the best-known positions in the search space that could satisfy the requirements of objective, targeted goals. It is worth noting that the proposed algorithm guarantees convergence to a global optimum, no matter how many candidate solutions are generated at the beginning.

## 4.3 Finding optimal solutions for multi-criteria decision-making

In the final iterations of each item, Pareto-front plotting (red line) presents convergence characteristics of the proposed algorithms at various item demands in a multi-objective function (Fig 10). A solution-possibility border (red line) illustrates a Pareto-efficient frontier, where the frontier and the region to its left and above comprise a continuous set of options. The points on the frontier show the Pareto-optimal options since there are locations on the border that is Pareto-dominated; points outside the frontier are not Pareto-efficient (blue dots). As described in Eq (15), the significance of objective functions is illustrated by selecting two coefficients $\mu$ and $\lambda$. The profit function finds the most significant value, whereas the area function gets the smallest value since the two objective functions are not optimally identical. The decision-maker clarifies the trade-offs between conflicting objectives, considering the relative importance of the targeted goals. In this case, they are equally $\mu = \lambda = 0.5$.

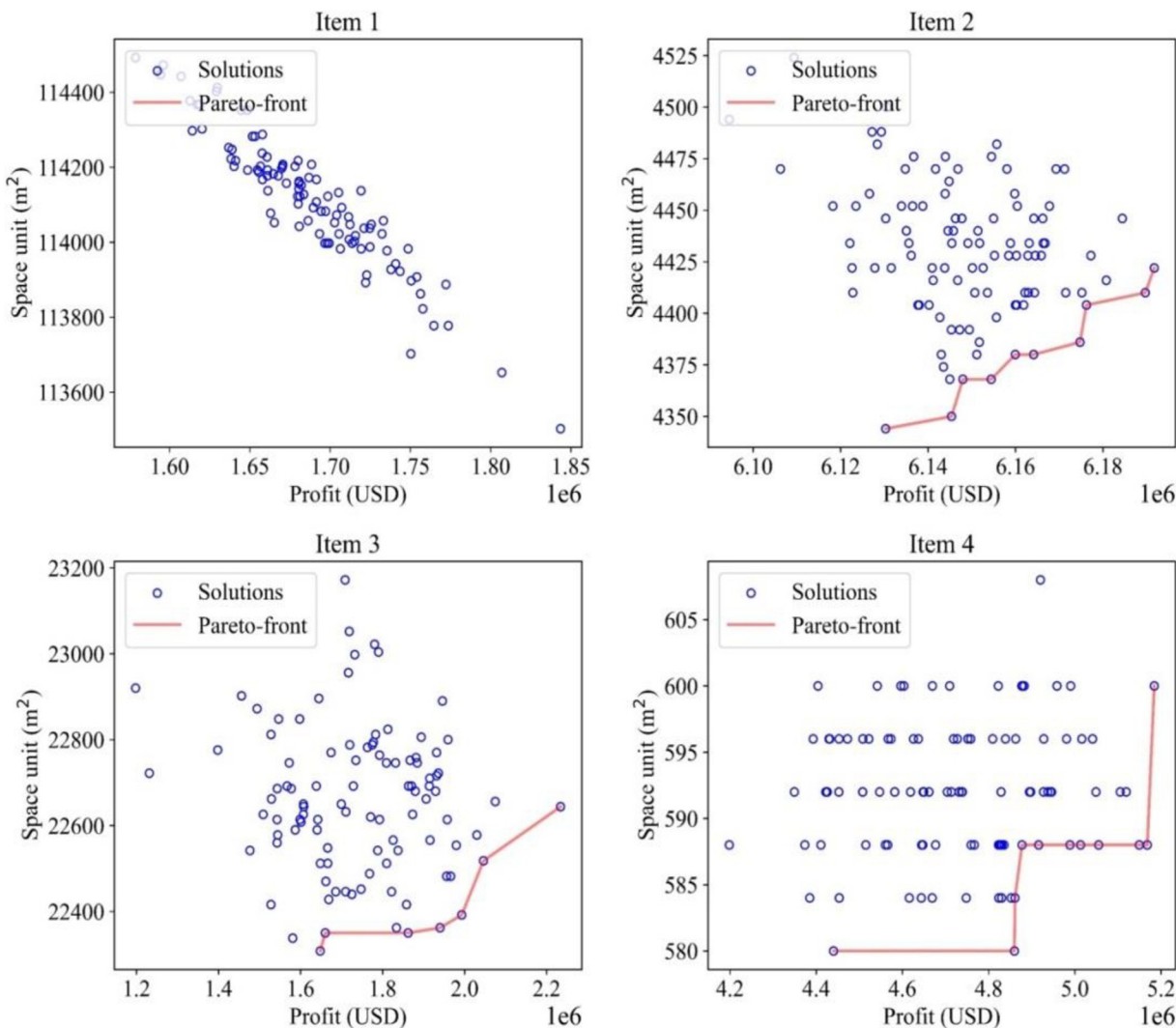

**Fig 10. Pareto frontier of the optimization process.**

In Fig 11, with two objective functions having equal weight, the optimal solution could be selected from a set of solutions obtained after optimization. The optimal solution is represented by a green "X" dot on the Pareto-front red line for each item. MOGWO explores possible solutions for each item. By applying the Pareto-front method, an optimum set could be found from group solutions. However, in item 1's solution, only one optimal solution could be achieved, and that solution is marked as a point. Therefore, this solution will always be optimum even if the weights of the objective functions change. In comparison, other items' solutions come with several points or solutions, which means that changing the function coefficients may affect the optimal solutions.

## 4.4 Sensitive analysis

The sensitivity analysis is intended to find out ranges of change where the optimal solution is sensitive to variable changes. The sensitivity analysis can specifically describe how slight changes in input parameter values affect the optimal inventory model. The sensitive analysis is

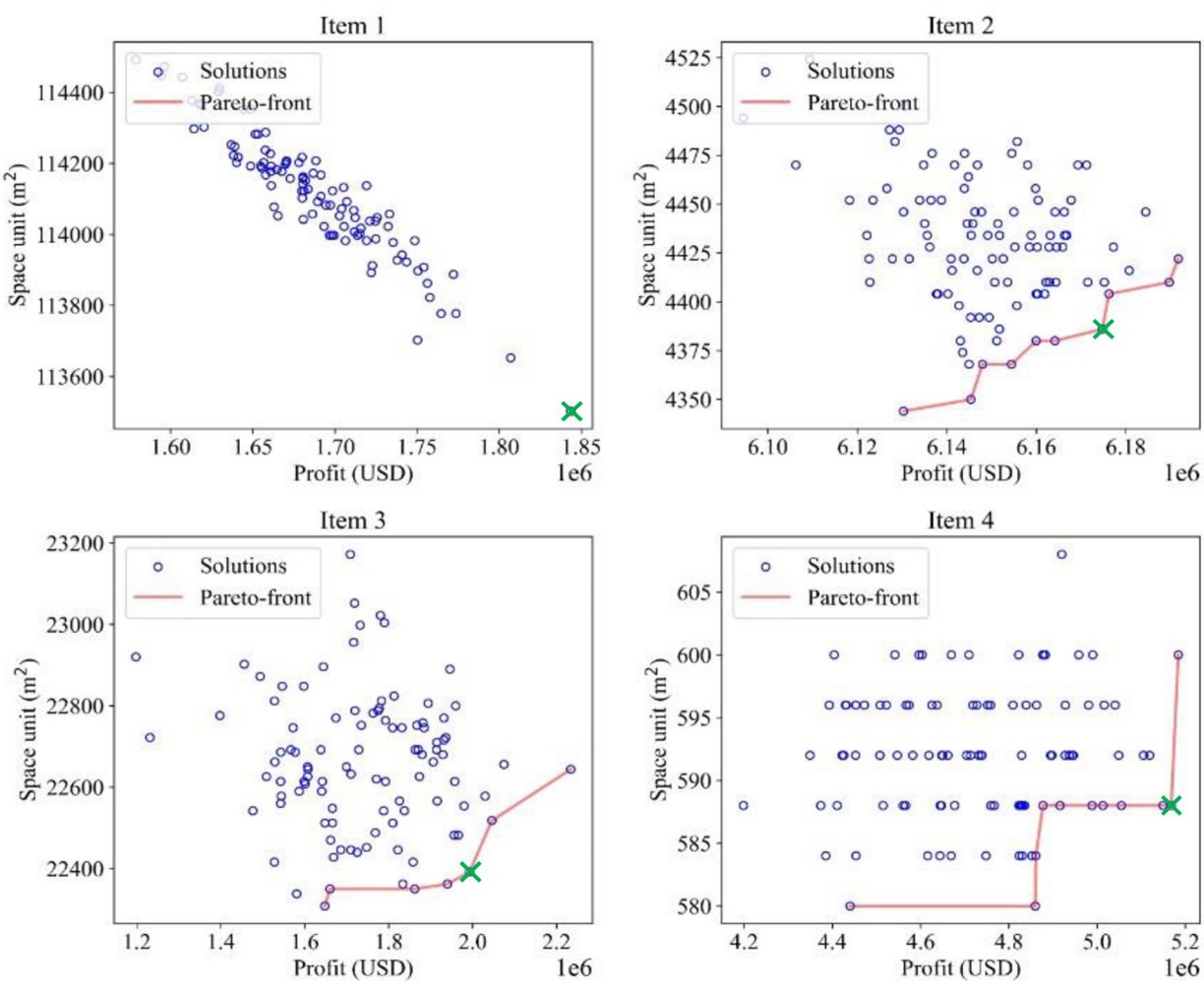

**Fig 11. Optimal solution from a set of solutions ($\mu = \lambda = 0.5$).**

presented in Table 5, representing the various objective values under the influence of order quantity ($Q$) and reorder point ($r$). By starting with item 1, when optimal values ($Q$, $r$) increase by 20% and 50%, profits values ($\Omega_\Sigma$) also decrease as follows, and storage space values ($\Pi_\Sigma$) tend to get larger. Meanwhile, reducing $Q$ and $r$ values by 20% and 50%, respectively, the values of the objective functions tend to change inversely as follows, the profit value ($\Omega_\Sigma$) rises, and the storage space value ($\Pi_\Sigma$) tends to grow less. This trend also applies to the remaining items. In conclusion, it can be argued that the connection between the storage space objective function ($\Pi_\Sigma$) and the choice variable ($Q$, $r$) is probably proportional. However, the objective function ($\Omega_\Sigma$) tends to be inversely proportional to the two values of the order quantity ($Q$) and reorder point ($r$). It is worthy of note that evaluating multiple scenarios facilitates understanding the sensitivity of outputs to specific inputs.

## 5. Conclusions and future works

This article presents an efficient inventory management model employing a multi-objective optimization framework under stochastic demand. The beginning inventory level in the prior period is different for each item, and the lead time in each unit varies. The multiple goals are

**Table 5. Sensitive analysis for inventory items ($\mu = \lambda = 0.5$).**

| Item | Changing parameter | Level of change | Profit ($\Omega_\Sigma$) (USD) | Storage space ($\Pi_\Sigma$) (m²) |
|---|---|---|---|---|
| Item 1 | $(Q, r)$ | -50% | 5985933 | 96230 |
| | | -20% | 5988136 | 96424 |
| | | 0% | 5928237 | 96553 |
| | | +20% | 5898088 | 96682 |
| | | +50% | 5863154 | 96875 |
| Item 2 | $(Q, r)$ | -50% | 6588470 | 1428 |
| | | -20% | 6462856 | 2284 |
| | | 0% | 6361578 | 2856 |
| | | +20% | 6314891 | 3427 |
| | | +50% | 6174229 | 4284 |
| Item 3 | $(Q, r)$ | -50% | 4807447 | 3486 |
| | | -20% | 4515990 | 5577 |
| | | 0% | 4150333 | 6972 |
| | | +20% | 4048255 | 8366 |
| | | +50% | 3873564 | 10458 |
| Item 4 | $(Q, r)$ | -50% | 4590670 | 5572 |
| | | -20% | 4630519 | 6126 |
| | | 0% | 4096278 | 6496 |
| | | +20% | 4008547 | 6865 |
| | | +50% | 3866354 | 7420 |

to maximize profit and minimize the required storage space. The study aims to determine each product's optimal order quantity and reorder point to optimize the objective functions with the constraints. The developed model is a nonlinear programming model mixed with binary variables, and the optimization algorithms have been realized to solve the inventory management problem.

The numerical tests are presented for evaluating multiple scenarios in four specific inventory items. Sensitivity analysis is performed to verify the optimal solutions further, proving the proposed algorithm's robustness and providing insights into how decision variables affect the optimal solutions. As the main contribution, the presented system offers a new paradigm to solve inventory problems by optimizing multi-objective problems involving profit and storage space. Test results demonstrate that the MOGWO algorithm can handle inventory problems with a significant difference in the range of values in objective functions. The MOGWO algorithm's superiority in terms of coverage and convergence has been proven through the leader selection strategy. To support these claims, this study presents extensive numerical tests and performs sensitivity analysis to verify the optimal solutions under stochastic demand. Implementing the MOGWO algorithm can significantly improve the efficiency of inventory management. Multi-objective optimization techniques can effectively help managers in complex decision-making processes.

However, a possible limitation should be considered for future work. A discount policy might be incorporated into the multi-objective optimization model, making the inventory management problem more realistic. Discount strategy can effectively reduce inventory costs by attracting new customers and increasing the sale of goods. In addition, the MOGWO algorithm can help decision-makers implement an optimal policy for order quantity and reorder point. Future studies can consider discount policy in the optimization model, employ actual data to explore practical marketing scenarios, and compare MOGWO and other algorithms.

Finally, the presented methodology can ensure more supply chain visibility and traceability under highly uncertain markets.

## Acknowledgments

This research was supported by "Regional Innovation Strategy (RIS)" through the National Research Foundation of Korea(NRF) funded by the Ministry of Education(MOE)(2023RIS-007).

## Author Contributions

**Conceptualization:** Nguyen Duy Tan, Sam-Sang You.

**Data curation:** Le Ngoc Bao Long, Duy Anh Nguyen.

**Formal analysis:** Le Ngoc Bao Long.

**Funding acquisition:** Hwan-Seong Kim, Sam-Sang You.

**Investigation:** Hwan-Seong Kim.

**Methodology:** Nguyen Duy Tan, Hwan-Seong Kim, Duy Anh Nguyen.

**Project administration:** Hwan-Seong Kim.

**Resources:** Hwan-Seong Kim.

**Software:** Nguyen Duy Tan, Le Ngoc Bao Long, Duy Anh Nguyen.

**Supervision:** Sam-Sang You.

**Validation:** Sam-Sang You.

**Visualization:** Le Ngoc Bao Long, Duy Anh Nguyen.

**Writing – original draft:** Le Ngoc Bao Long.

**Writing – review & editing:** Sam-Sang You.

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
