## [Decision Letter · Decision Letter 0]

5 Feb 2023

PONE-D-23-01150Optimization and inventory management under stochastic demand using metaheuristic algorithmPLOS ONE

Dear Dr. You,

Thank you for submitting your manuscript to PLOS ONE. After careful consideration, we feel that it has merit but does not fully meet PLOS ONE’s publication criteria as it currently stands. Therefore, we invite you to submit a revised version of the manuscript that addresses the points raised during the review process.

We look forward to receiving your revised manuscript.

Kind regards,

Seyedali Mirjalili

Academic Editor

PLOS ONE

Journal Requirements:

"This research was supported by Korea Institute of Marine Science & Technology Promotion (KIMST) funded by the Ministry of Oceans and Fisheries, Korea (20220573)"

Reviewers' comments:

Reviewer's Responses to Questions

**Comments to the Author**

1. Is the manuscript technically sound, and do the data support the conclusions?

Reviewer #1: No

Reviewer #2: Partly

2. Has the statistical analysis been performed appropriately and rigorously? 

Reviewer #1: Yes

Reviewer #2: No

3. Have the authors made all data underlying the findings in their manuscript fully available?

Reviewer #1: Yes

Reviewer #2: Yes

4. Is the manuscript presented in an intelligible fashion and written in standard English?

Reviewer #1: No

Reviewer #2: Yes

5. Review Comments to the Author

Reviewer #1: This manuscript considers multi-period inventory systems for optimizing profit and storage space under stochastic demand in supply chain management. A nonlinear programming model based on random demand is proposed to simulate the inventory operation. . Adequate revisions to the following points should be undertaken to justify the recommendation for publication.

The abstract section is fragile. Please re-write an abstract section, explain an obtained result and contribution, improve a proposed method, etc. Please delete unnecessary information.

This paper has more than spelling and grammatical errors. Please fix all of them.

The authors should clearly state the limitations of the proposed method in other applications.

There are many multi-objective GWO algorithms in the literature, what is the difference between your algorithm with theirs?

Please add a flowchart of the proposed method.

The related work section is missing; please add this section. And make the Introduction and related work sections more productive using the following articles. Reading and using these articles:

o https://doi.org/10.1007/s00354-022-00188-w

o https://doi.org/10.1007/s42235-022-00288-9

o https://doi.org/10.1007/s00366-021-01369-9

References are not up to date and not enough. Please improve this item.

How did the authors set parameters for your proposed algorithm? Please make sensitivities of these parameters to the performance of your proposed algorithm!

All the structural problems in this study are very popular, therefore the authors are suggested to compare with results obtained from previous studies to convince the power of the proposed algorithm. Moreover, these structural problems are very easy, please try to solve other complex problems.

The authors should provide the code of your proposed algorithm.

Please write a contribution to your paper in the Introduction section.

Please change the title of the end section (Conclusion) to (Conclusion and Future Works), and write some future works.

Please use a new comparison algorithm, such as the Farmland fertility algorithm, African Vultures Optimization Algorithm, Mountain Gazelle Optimizer, and Artificial Gorilla Troops Optimizer.

Expand the critical results in the conclusion. Focus on the main developments in the finale. Also, write the main contributions in the conclusion.

Numerical results are good enough, but more explanations are required to analyze each figure presented.

All figures have low quality, and please improve all of them.

Good luck

Reviewer #2: 1. The novelty and findings of this study are not stated in the Abstract, and it is recommended to rewrite it.

2. The English language needs significant revision.

3. The title cannot reflect this problem and novelty, and it is recommended to revise it.

4. The contributions of this study should be summarized at the end of the Introduction.

5. This study suffers from a lack of an in-depth literature review on the problem of this study and recent metaheuristic algorithms through which reviewing the following recent algorithms are recommended. Quantum-based avian navigation optimizer algorithm, An improved moth-flame optimization algorithm with adaptation mechanism to solve numerical and mechanical engineering problems, DMDE: Diversity-maintained multi-trial vector differential evolution algorithm for non-decomposition large-scale global optimization, Migration-based moth-flame optimization algorithm, and Starling murmuration optimizer: A novel bio-inspired algorithm for global and engineering optimization.

6. The nomenclature table is recommended for the parameters used in all equations.

7. The importance of this study is not clear why the new multi-objective version of GWO is proposed.

8. Please check the title "3. Multi-objective optimization using grey wolf optimizer".

9. This study's novelty is unclear, and it is suggested to consider the proposed method section and explain your novelty clearly.

10. The claims of this study should be stated, and the authors explain which experimental evaluation can support their claims.

11. It is recommended to check the pseudocode of the multi-objective grey wolf optimizer (MOGWO).

12. It is recommended to compare MOGWO with other multi-objective algorithms.

13. The Conclusion should be extended to include more details regarding the limitations of the proposed study.

14. The nonparametric statistical test is recommended.

6. PLOS authors have the option to publish the peer review history of their article (what does this mean?). If published, this will include your full peer review and any attached files.

Reviewer #1: No

Reviewer #2: No

---

## [Author Response · Author response to Decision Letter 0]

30 Mar 2023

Please check the attached file "23Plone_tanAA_QA_Fnn" for details.

---

## [Decision Letter · Decision Letter 1]

2 May 2023

PONE-D-23-01150R1Optimization and inventory management under stochastic demand using metaheuristic algorithmPLOS ONE

Dear Dr. You,

Thank you for submitting your manuscript to PLOS ONE. After careful consideration, we feel that it has merit but does not fully meet PLOS ONE’s publication criteria as it currently stands. Therefore, we invite you to submit a revised version of the manuscript that addresses the points raised during the review process.

We look forward to receiving your revised manuscript.

Kind regards,

Seyedali Mirjalili

Academic Editor

PLOS ONE

Reviewers' comments:

Reviewer's Responses to Questions

**Comments to the Author**

1. If the authors have adequately addressed your comments raised in a previous round of review and you feel that this manuscript is now acceptable for publication, you may indicate that here to bypass the “Comments to the Author” section, enter your conflict of interest statement in the “Confidential to Editor” section, and submit your "Accept" recommendation.

Reviewer #1: (No Response)

Reviewer #2: (No Response)

2. Is the manuscript technically sound, and do the data support the conclusions?

Reviewer #1: (No Response)

Reviewer #2: Yes

3. Has the statistical analysis been performed appropriately and rigorously? 

Reviewer #1: (No Response)

Reviewer #2: N/A

4. Have the authors made all data underlying the findings in their manuscript fully available?

Reviewer #1: (No Response)

Reviewer #2: Yes

5. Is the manuscript presented in an intelligible fashion and written in standard English?

Reviewer #1: (No Response)

Reviewer #2: Yes

6. Review Comments to the Author

Reviewer #1: From the response letter, I think the paper has been well revised according to the previous reviewers, and the current version of the manuscript is acceptable for publication.

Reviewer #2: Thank the authors for responding to some of the comments; however, they have not responded to some of the suggested comments adequately. I recommend authors address the following issues before accepting for publication.

1. It is recommended to revise a paper based on the journal's guidelines. The style of references should be revised, and some sentences and abbreviations of the algorithm need references. Please check the following paragraph. "Lately, the following approaches have attracted extensive attention in solving various problems owing to their…"

2. The realted work still can be improved by adding more recent algorithms as well as their applications.

3. In Fig. 4, the arrows of boxes t+1 and the calculated fitness value should be rechecked.

4. The description of equations 15 and 16 should be revised.

5. The claims of this study should be stated, and the authors explain which experimental evaluation can support their claims.

7. PLOS authors have the option to publish the peer review history of their article (what does this mean?). If published, this will include your full peer review and any attached files.

Reviewer #1: No

Reviewer #2: No

---

## [Author Response · Author response to Decision Letter 1]

10 May 2023

Please check the attached file for Respond to Reviewers.

---

## [Decision Letter · Decision Letter 2]

16 May 2023

Optimization and inventory management under stochastic demand using metaheuristic algorithm

PONE-D-23-01150R2

Dear Dr. You,

We’re pleased to inform you that your manuscript has been judged scientifically suitable for publication and will be formally accepted for publication once it meets all outstanding technical requirements.

Kind regards,

Seyedali Mirjalili

Academic Editor

PLOS ONE

Additional Editor Comments (optional):

Reviewers' comments:

Reviewer's Responses to Questions

**Comments to the Author**

1. If the authors have adequately addressed your comments raised in a previous round of review and you feel that this manuscript is now acceptable for publication, you may indicate that here to bypass the “Comments to the Author” section, enter your conflict of interest statement in the “Confidential to Editor” section, and submit your "Accept" recommendation.

Reviewer #2: All comments have been addressed

2. Is the manuscript technically sound, and do the data support the conclusions?

Reviewer #2: Yes

3. Has the statistical analysis been performed appropriately and rigorously? 

Reviewer #2: N/A

4. Have the authors made all data underlying the findings in their manuscript fully available?

Reviewer #2: Yes

5. Is the manuscript presented in an intelligible fashion and written in standard English?

Reviewer #2: Yes

6. Review Comments to the Author

Reviewer #2: The authors have responded to most of the comments, and I recommend accepting the revised manuscript for publication after careful checking and correcting some minor changes, for example:

- "QANO" should be changed to "QANA".

- "Sahoo et al. [15] presented the Migration-based ..." should be changed to "Nadimi-Shahraki et al. [15] presented the Migration-based ...".

- Reference number 15, referring to the M-MFO algorithm, should be corrected.

7. PLOS authors have the option to publish the peer review history of their article (what does this mean?). If published, this will include your full peer review and any attached files.

Reviewer #2: No

---

## [Editor Report · Acceptance letter]

10 Jul 2023

PONE-D-23-01150R2 

Optimization and inventory management under stochastic demand using metaheuristic algorithm 

Dear Dr. You:

I'm pleased to inform you that your manuscript has been deemed suitable for publication in PLOS ONE. Congratulations! Your manuscript is now with our production department. 

Kind regards, 

on behalf of

Prof. Seyedali Mirjalili 

Academic Editor

PLOS ONE